# Preparation of Flame-Retardant Polyurethane and Its Applications in the Leather Industry

**DOI:** 10.3390/polym13111730

**Published:** 2021-05-25

**Authors:** Shaolin Lu, Yechang Feng, Peikun Zhang, Wei Hong, Yi Chen, Haojun Fan, Dingshan Yu, Xudong Chen

**Affiliations:** 1Key Laboratory for Polymeric Composite and Functional Materials of Ministry of Education, School of Chemistry, Sun Yat-Sen University, Guangzhou 510275, China; lushlin5@mail.sysu.edu.cn (S.L.); fengych9@mail2.sysu.edu.cn (Y.F.); hongwei9@mail.sysu.edu.cn (W.H.); yudings@mail.sysu.edu.cn (D.Y.); 2Key Laboratory of Leather Chemistry and Engineering of Ministry of Education, College of Biomass Science and Engineering, Sichuan University, Chengdu 610065, China; houdelong@stu.scu.edu.cn (P.Z.); chenyi_leon@scu.edu.cn (Y.C.)

**Keywords:** flame-retardant, polyurethane, synthetic-leather

## Abstract

As a novel polymer, polyurethane (PU) has been widely applied in leather, synthetic leather, and textiles due to its excellent overall performance. Nevertheless, conventional PU is flammable and its combustion is accompanied by severe melting and dripping, which then generates hazardous fumes and gases. This defect limits PU applications in various fields, including the leather industry. Hence, the development of environmentally friendly, flame-retardant PU is of great significance both theoretically and practically. Currently, phosphorus-nitrogen (P-N) reactive flame-retardant is a hot topic in the field of flame-retardant PU. Based on this, the preparation and flame-retardant mechanism of flame-retardant PU, as well as the current status of flame-retardant PU in the leather industry were reviewed.

## 1. Introduction

Polyurethane (PU) refers to a polymer with repeating carbamate groups (–NHCOO–) in the polymer backbone structure [1]. PU was first synthesized from hexamethylene diisocyanate and 1, 4-butanediol [2]. PU has been highly appreciated by both academia and industry for its excellent performance and has been regarded as the “fifth most common plastic” after polyethylene, polyvinyl chloride, polypropylene, and polystyrene [3]. To date, a wide range of PU products has played a pivotal role in many fields and contributed to the improvement of people’s everyday lives [4,5,6].

Based on its diversity, PU has become the polymer that exhibits a wide range of application in both physical and chemical properties [7]. Currently, various PU products have been widely applied in leather fabrication, synthetic leather, the light industry, textiles, coatings, building materials, electronics, medicine, automotive, national defense, and aerospace (see Figure 1). Indeed, PU has become one of the most versatile and fastest growing high-tech materials in modern society [7,8,9,10,11].

As one of the most commonly used chemicals in the leather industry, PU is mainly used as a finishing agent and retanning filler [12,13,14,15]. In PU synthetic leather, the PU, substrate, and most additives are flammable or combustible materials; therefore, the LOI of synthetic leather products is generally less than 21% and defined as a combustible material. When burned, PU releases a high concentration of HCN, HNCO, CO, NO, NO_2_, and other hazardous gases and fumes [16], which are a great safety hazard threatening people’s lives and property. With improved standards of living around the world, the demand for high-performance and high-quality PU products continues to rise. Therefore, improving the heat resistance and flame retardancy of PU materials is not only of great theoretical significance, but also of high practical value. Currently, the main methods of PU flame retardancy are blending and addition, reaction grafting, and nano-composites. This article reviews the preparation method and flame-retardant mechanism of flame-retardant PU, starting from the synthesis and structural characteristics of PU. In addition, the current application status of flame-retardant PU in the leather industry is summarized, along with the problems it currently faces.

## 2. Synthesis and Structural Characteristics of PU

The synthesis of PU is based on the chemical reaction of isocyanates, which is obtained by the stepwise addition polymerization of basic raw materials (e.g., polyisocyanate and compounds containing active hydrogen). Currently, industrial PU is produced by the reaction of diisocyanate and polyols (e.g., polyether or polyester polyols, small molecule chain expanders, and/or crosslinkers) (see Figure 2). From the perspective of molecular structure, PU is a typical block polymer in which the macromolecular chains are mainly composed of alternating flexible (soft) and rigid (hard) chain segments. Among them, the soft segment formed by the macromolecule polyols is flexible and presents a random curl; the carbamate group formed by the isocyanate and small molecule chain extender is a rigid segment, which has a relatively large cohesion energy and can be easily aggregated and stretched into a rod shape. Due to the incompatibility of the soft and hard segments, there is micro-phase separation in PU, which affects the physical properties of PU materials.

The molecular structure of PU has high tunability through the selection of different types of raw materials, adjusted ratios of monomers, reaction sequence control, and the use of different synthetic process conditions. It can be prepared to generate a variety of structures, morphologies, processing methods, and appearances and media, including PU soft foam, PU rigid foam, PU elastomer, PU fibers, PU coatings, and PU adhesives [7,8,9,10,17].

## 3. Progress of Flame-Retardant PU

The flammability of PU is a great safety hazard, threatening lives and possessions. Therefore, the flame-retardant modification of PU has attracted the attention of researchers and enterprises. Currently, the primary methods imparting flame retardancy include adding flame-retardant additives to PU material through blending, grafting, and nano-compounding. In the meantime, we briefly discuss the potential health risks of the flame-retardant polyurethane to human beings.

### 3.1. Additive Flame-Retardant PU

Additive flame-retardant PU is to add a flame-retardant to PU resin by physically compounding to reduce the flammability of PU or even achieving non-flammability. Commonly used additive flame retardants include inorganic flame retardants or organic compounds containing halogens, such as phosphorus and nitrogen. The flame-retardant mechanism of inorganic flame retardant is decomposition decalescence and crystal water release. The formed metal oxide particles can be used as a barrier and inhibit the flow of polymer melt, which has a significant suppression effect on the melt drips [18]. In condensed phase, P-containing compounds mediate the formation of char residues mainly through inducing cyclization, cross-linking and aromatization/graphitization by dehydration of polymer structure. In the process of hydrocarbon combustion in gas phase, P-containing compounds decompose to form P·radicals and react with OH· radicals, which makes fuel combustion cycle “flame poisoning” [19]. The typical flame-retardant mechanism of nitrogen-based flame retardants is that a large number of non-combustible gases are produced in the decomposition process, which can prevent the combustion of polymers by diluting the oxygen concentration [20]. Currently, there are several additive flame retardants available in the market (see Table 1) that can impart good flame retardancy to PU. For instance, Shi et al. investigated the effects of combining dimethyl methylphosphonate (DMMP) and tris(2-chloroethyl) phosphate (TCEP), tris(2-chloroisopropyl) phosphate (TCPP), and tris(1, 3-dichloroisopropyl) phosphate (TDCP) on the oxygen index and combustion performance of urethane rigid foams. The results demonstrated that DMMP and TDCP showed the best performance [21]. Price et al. demonstrated that melamine (MA) was effective in reducing the peak heat and smoke release rates of PU during combustion [22]. Chen et al. investigated the effects of the ammonium polyphosphate/iron oxyhydroxide (APP/FeOOH) system on thermoplastic PU (TPU) flame retardancy. The results demonstrated that when 18.75% of APP and 1.25% of FeOOH were added, the ultimate oxygen index (LOI) of TPU reached 32.8% and the total heat and smoke release of TPU was significantly reduced [23].

In addition to the above commercially available additive flame retardants, researchers have also prepared PU additive flame retardants containing phosphorus, nitrogen, and other flame-retardant elements via organic synthesis [32]. Among them, phosphorus-containing and P-N flame retardants have been reported numerous times. For instance, Gaan et al. synthesized a series of phosphonate, phosphate, and phosphoramidite flame retardants (see Figure 3a). The results demonstrated that phosphonates and phosphoramide-like flame retardants could generate more ∙PO_2_ and ∙PO free radicals in the gas phase, resulting in higher flame-retardant efficiency than phosphate flame retardants [33,34]. Then, Gaan et al. also developed a series of DOPO (9, 10-dihydro-9-oxa-10-phospha-phenanthrene-10-oxide) phosphoramidite flame retardants (chemical structures of PA-DOPO, BA-DOPO and EDAB- DOPO were shown in Figure 3b), which exhibited better flame retardancy than commercial flame-retardant TCPP in PU foam [35]. König et al. prepared a methylated DOPO flame-retardant (see Figure 3c), which was able to self-extinguish PU foam with only 7.5% flame-retardant [36]. Wang et al. synthesized Schiff base polyphosphate flame-retardant (SPE, see Figure 3d) and applied it to TPU and was able to achieve 29% LOI of TPU with 5% Schiff-base polyphosphate ester (SPE) and passed the UL-94 V-0 test [37]. Wang et al. prepared an intumescent flame-retardant MATMP (see Figure 3e) from melamine (MA) and amino-trimethylene phosphonic acid (ATMP), and the oxygen index of the flame-retardant PU was 25.5% at a mass fraction of 15% of MATMP, and the flame-retardant PU achieved UL-94 V-0. When the mass fraction of MATMP was 15%, the oxygen index of flame-retardant PU was 25.5%, and it reached UL-94 V-0 level [38]. In addition, Wang et al. synthesized a one-component intumescent flame-retardant DPPM containing MA group (see Figure 3f). The results demonstrated that the addition of 25% DPPM could enhance the oxygen index of flame-retardant PU to 29.5% [39].

Expandable graphite (EG), it should be noted, has been widely applied in polymer as flame retardants, especially in rigid polyurethane foam (RPUF) [24,25,40,41,42,43], due to its low toxic, abundance, and high flame-retardant efficiency. The flame-retardant mechanism of EG is that it can immediately expand (150~300 times) at relatively low temperature, and then form a worm-like protective char layer in the condensed phase, which is an effective barrier to prevent heat and mass transfer [24,42,43]. However, the loose “graphite worm” residues are easy to collapse, which hinders the further improvement of the flame-retardant efficiency of EG. Therefore, EG was usually used with other flame retardants, such as inorganic hydrated compounds [24,25], P/N/Si-containing organic compounds [40,41,42,43].

Although the research and development of additive flame retardants is flourishing [44,45,46] and can impart good flame retardancy to PU, unfortunately, due to the low flame-retardant efficiency of additive flame retardants and its poor compatibility with the PU matrix, the mechanical performance of PU often rapidly deteriorates and the flame retardant is prone to migrate and precipitate during use [47]. These defects severely restrict the further application of additive flame retardants. Therefore, how to improve PU flame retardancy while accounting for the excellent performance of the PU matrix itself is a major challenge in the field of flame-retardant PU. In recent years, more researchers have focused on reactive flame-retardant PU systems.

### 3.2. Reactive Flame-Retardant PU

Reactive flame retardancy, also known as intrinsic flame retardancy, is obtained by introducing polyols-containing flame-retardant elements (e.g., halogen, phosphorus, nitrogen, silicon, boron), isocyanates, chain extenders, or curing agents into the PU molecular chain through chemical bonding, to achieve the flame retardancy of PU. Based on the differences of flame-retardant elements, reactive flame-retardant PU can be divided into halogen-containing, nitrogen-containing, phosphorus-containing, and P-N flame-retardant PU.

#### 3.2.1. Halogen Reactive Flame-Retardant PU

Halogen flame retardants have a good gas-phase flame-retardant effect, high flame-retardant efficiency, and low cost, and was the first type of flame-retardant used in flame-retardant PU. Halogen reactive flame-retardant PU is realized by designing halogen-containing polyols through organic synthesis and then reacting with isocyanate to obtain flame-retardant PU [48]. For instance, Solvay’s halogenated polyols Ixol M125 contains 32% and 6.8% bromine and chlorine elements, respectively, and the oxygen index of PU made from it can reach 30% [49]. In addition, Park et al. synthesized polyester diols with chlorine and bromine side chains, respectively. The results demonstrated that the prepared flame-retardant PU failed to ignite in vertical combustion tests and passed the UL-94 V-0 test [50,51]. Although halogenated flame retardants can provide PU with high flame-retardant efficiency and are inexpensive, halogenated flame retardants emit large amounts of smoke when burned and produce corrosive and toxic gases such as hydrogen halides, dioxins, and furans. Additionally, it has been shown to accumulate and enrich in living organisms, posing a serious risk to human health and environmental safety [52]. Therefore, the prospect of halogen-containing polyols has been overshadowed. Specifically, with the promulgation of EU RoHS (Restriction of Hazardous Substances) and WEEE (Waste Electrical and Electronic Equipment) directives, halogen-containing flame retardants have been pushed to the forefront and the development of new methods for halogen-free flame-retardant PU is the most popular trend.

#### 3.2.2. Nitrogen Reactive Flame-Retardant PU

The synthesis of nitrogen-containing flame-retardant polyols is usually based on isocyanurate ring compounds and melamine as starting agents. For instance, Rotaru et al. used 1, 3-oxazolidine and isocyanuric acid as raw materials to synthesize polyols-containing isocyanurate heterocyclic structures by the Mannich reaction, and the obtained PU showed good flame retardancy [53]. Zhang et al. also used cashew phenol and MA as raw materials to synthesize a Mannich base polyol containing an imino-triazine ring structure by the Mannich reaction. The results demonstrated that flame retardancy of PU rigid foam prepared using this polyol could be improved [54]. Hu et al. prepared melamine-based polyols (MADP) containing six active hydroxyl groups from MA, which could enhance both the thermal stability and flame retardancy of PU rigid foam [55]. Nevertheless, there is little research on nitrogen-reactive flame retardants, as flame retardancy of PU prepared using only nitrogen-containing flame retardants is rather limited. Therefore, nitrogen-reactive flame retardants are typically used in conjunction with other systems.

#### 3.2.3. Phosphorus Reactive Flame-Retardant PU

Unlike halogenated flame retardants, phosphorus flame retardants emit less hazardous gases and fumes during combustion, and since phosphorus flame retardants exhibit flame-retardant effects in both the solid and gas phases, they are more effective for PU [56,57]. Phosphorus-containing polyols are the main phosphorus-reactive flame retardants for PU. Phosphorus-containing polyols are generally composed of phosphides, polyhydroxy compounds, and epoxides. Depending on the oxidation state of phosphorus, phosphorus-reactive flame retardants can be further classified into phosphine compounds, phosphine oxide, organophosphates, phosphonate, and phosphite [32]. Chen et al. used Clariant’s Exolit OP550 diol (see Figure 4a) to modify WPU (waterborne polyurethane), and a 15% usage rate could increase the oxygen index of flame-retardant WPU to 33.1% [58]. Nevertheless, the thermal stability of the phosphate ester flame-retardant PU matrix tends to decrease due to the heat degradation of the phosphate ester groups [59]. Given the higher thermal stability of the P-C bond compared to the P-O bond [60], preparation of phosphonate and phosphorus oxide-reactive polyols have been widely reported. For instance, Chiu et al. used polycaprolactone diols (PCL) and phosphonate diols HMCPP (see Figure 4b) as soft segments to produce PU with a UL-94 V-0 rating [61]. Hu et al. synthesized phosphonate diols BHPP (see Figure 4c), and the PU rigid foam prepared using them as chain extenders showed good flame retardancy [62]. All phosphonate polyols showed good flame retardancy for PU [63,64,65]. To further enhance the thermal stability of phosphorus polyols, Wang et al. synthesized a trihydroxy phosphine oxide (see Figure 4d) and used it as a cross-linking agent to obtain intrinsically flame-retardant PU [66]. Lee et al. reported that phosphine oxide-containing diols FR-D (see Figure 4e) could also impart good flame retardancy to PU-rigid foam [67].

Notably, at present, most of the reported phosphate esters and phosphonate flame retardants have P-O bonds located in the main chain of the molecule, and the P-O bonds in the molecular structure have problems of being readily hydrolyzable. The hydrolysis of the P-O bond in the molecular structure will inevitably lead to the breakage of the main chain structure of PU, which will destroy the mechanical performance of the substrate and greatly shorten the service life of the material while leading to potential hazards [68,69]. To address this problem, Fan et al. designed a flame-retardant diol, EPPD, with phosphorus-containing side chains (see Figure 4f). The results demonstrated that by “suspending” the P-O bond in the PU side chain, the hydrolysis of the phosphoric acid ester could be effectively slowed and the damage to the main chain of WPU could be avoided even if the WPU emulsion was subject to elevated temperature [70]. On the other hand, the efficiency of phosphorus-containing flame-retardant PU systems is limited, and to achieve the desired flame retardancy, it is often necessary to sacrifice key properties (especially the mechanical performance) of the PU matrix itself. Hence, more researchers are focusing on synergistic frame-retardant PU systems.

#### 3.2.4. P–N Reactive Flame-Retardant PU

The introduction of both phosphorus and nitrogen flame-retardant elements into the PU structure can achieve better flame retardancy by a synergistic P-N flame-retardant effect [71]. Based on this, the development and utilization of nitrogen and phosphorus reactive flame retardants has become a hot topic in the research of flame-retardant PU materials in recent years.

In the early 21st century, the development of P-N reactive flame retardants for PU was primarily based on curing agents. For instance, Chen et al. synthesized a variety of phosphorus-containing curing agents with aziridinyl groups (see Figure 5a) and cured them with WPU to obtain PU with good flame retardancy [72,73,74,75]. Chen et al. synthesized a methacryloyloxyethyl-terminated curing agent NPHE with cyclophosphazene as the core from phosphonitrilic chloride trimer and hydroxyethylmethacrylate (see Figure 5b) and cured it with PU-acrylate prepolymer under UV irradiation to prepare flame-retardant PU [76]. Hu et al. prepared a P-N curing agent N-PBAAP containing a bis(acryloyloxy) group (see Figure 5c), and the cured PU also exhibited good flame retardancy [77]. Nevertheless, curing of the P-N curing agent with PU prepolymer needs to be carried out under specific conditions (e.g., UV light, high temperature heating), which requires high processing environment and equipment, and the operation is tedious, thus limiting its practical application in many fields.

The use of P-N flame-retardant as a reaction monomer (e.g., macromolecule polyols, small molecule chain extender or isocyanate) in the synthesis of PU to obtain intrinsic flame-retardant PU has been shown to be a simpler and more efficient method [19]. For instance, Ding et al. reacted P-N diols with methylene diphenyl diisocyanate (MDI) to obtain flame-retardant diisocyanate FRPUP (see Figure 6a), and prepared flame-retardant PU samples [78]. Liu et al. synthesized a P-N flame-retardant bis(4-isocyanatophenyl)phenylphosphine oxide (BIPPO, see Figure 6b), and the oxygen index of PU prepared with these polyols could reach 29–33% [79]. Although P-N isocyanate can impart flame retardancy to PU, there were some inherent defects, such as the -NCO group being easily reacted with water, which is not conducive to long-term storage. Therefore, P-N flame-retardant macromolecule polyols and small molecule chain expanders have become more popular among researchers.

Çelebi et al. synthesized the flame-retardant BAPPO (see Figure 6c) and substituted ethylenediamine as a chain extender, resulting in a PU LOI of 27% [80]. Wu et al. reported phosphoramidite diols (EMSPB, see Figure 6d) with a double spiro ring structure, and the LOI of PU rigid foam prepared with 25% EMSPB was 27.5%, while the flame-retardant level was UL-94 V-0 [81]. Recently, Wang et al. synthesized P-N macromolecular polyester diol (DMOP, see Figure 6e) by transesterification using DMMP and diethanolamine as the raw materials. The PU foam was prepared using only 6.3% DMOP and could self-extinguish in a vertical combustion test. At the same time, DMOP has excellent migration resistance, and the flame retardancy remained almost unchanged after 64 h at 140 °C [82]. The chemical structure and flame-retardant parameters of the corresponding flame-retardant polyols with built P and N atoms are listed in Table 2.

With the development of WPU in the 21st century, P-N reactive flame-retardant WPU has been a hot research topic in the field of flame-retardant PU over the past decade or two. Table 3 shows the chemical structures of typical P-N reactive flame retardants and their performance of flame-retardant WPU. For instance, Xu et al. used industrial P-N diols Fyrol-6 as a chain extender, and the LOI of WPU reached 29% when the phosphorus content of the system was 2.31% [86]. Luo et al. synthesized a phosphoramide flame-retardant ODDP with a ten-membered ring structure, and the WPU obtained by using it as a chain extender can pass UL-94 V-0 [87]. Fan et al. reported that a P-N intumescent flame-retardant BSPB with a diamino-containing double spiro structure embedded in the WPU backbone could improve the LOI of WPU to 27% using only 8% BSPB, which improved the char-forming and anti-dripping of WPU [88].

Currently, the development of reactive flame retardants based on P-N systems is the focus of research in the field of flame-retardant PU. Generally, the introduction of 10–20% of flame-retardant into the PU system can give better PU flame retardancy. Nevertheless, most of the reported P-N reactive flame retardants are phosphonates or phosphonate compounds, and the P-O bond in the molecular structure is located in the main chain. As mentioned earlier, the P-O bond is susceptible to deliquescence when exposed to water, which can damage the main chain structure of the PU matrix and affect its service life, especially in WPU where water is used as the dispersion medium. In addition, the low thermal stability of the P-O bond cannot be neglected. Hence, how to improve the flame retardancy of PU while reducing the impact on key properties (e.g., tensile strength, elongation at break, thermal stability, hydrolytic stability) is still a challenge in the field of flame-retardant PU that must be quickly addressed.

### 3.3. Flame-Retardant PU Nano-Composites

Since polymer/montmorillonite nano-composites were reported in the late 20th century, nano-composites have become a new topic in research and industry. To date, PU/nano-composite systems have been widely and intensively studied and are considered to be the best alternative to traditional flame retardants. To date, nanoparticles with proven flame-retardant effects mainly include carbon nanotubes (CNT) [92], fullerene (C_60_) [93], montmorillonite (MMT) [94], layered double hydroxides (LDH) [95], layered silicate (LS) [96], graphitic carbon nitride (CN) [97,98], and polyhedral oligomericsilsesquioxane (POSS) [99]. In general, a low loading of nanoscale flame-retardant (typically below 5 wt% [20]) can significantly improve the flame retardancy of PU composites. Meanwhile, unlike conventional flame retardants, nanoscale flame retardants show little effect on the mechanical performance of PU matrix and can even effectively enhance the mechanical performance of composites. This provides new ideas for the research and development of flame-retardant PU.

Since its discovery in 2004 [100], graphene has been a highly studied nanomaterial. Owing to its unique single layer, 2D nanoscale structure, graphene exhibits excellent electrical conductivity (6000 S∙cm^−1^), thermal conductivity (5000 W∙m^−1^∙K^−1^), and mechanical performance (tensile strength = 130 GPa, Young’s modulus = 1000 GPa). As a result, graphene has been widely applied in sensors, energy storage, coating, pollution treatment, and polymer composites [101,102]. Indeed, since graphene has a two-dimensional structure similar to that of LDH, CN, and LS, it can exert a “lamellar barrier effect” and thus shows promise in the field of flame retardancy. Second, graphene has a large specific surface area, which can also provide catalytic or carbonization platform for other materials such as metal oxides. Third, graphene contains rich active oxygen groups (carboxyl groups on the edge, epoxy and hydroxyl groups on the basal planes), which can undergo decomposition and dehydration at relatively low temperature, thus absorbing heat during combustion to cool the polymer matrix [103]. Shi et al. showed that graphene obtained by reduction of graphene oxide (GO) could not be ignited in air and only turned red when treated with high temperature flame, indicating its thermal stability and flame retardancy. This lays the foundation for further applications of graphene in flame-retardant materials [104]. For this reason, graphene has made great progress in epoxy resin (ER), polypropylene (PP), polystyrene (PS), polycarbonate (PC), polylactic acid (PLA), and other flame-retardant nano-composites [103,105].

It is worth noting that graphene has also been used in the field of flame-retardant PU, but there have been drawbacks. For instance, the flame-retardant efficiency of composites prepared by GO alone is low, and some studies have even shown that GO with a high degree of oxidation is used as a “fuel” in the polymer to promote the combustion of composites [106]. For this reason, Chen et al. added traditional flame retardants, such as graphene and ammonium polyphosphate [107], the melamine salt of pentaerythritol phosphate [108] to TPU resin by melt blending. The results demonstrated that graphene could promote char formation in TPU and passed the UL-94 V-0 test. Gavgani et al. used graphene as the carbon source, MA as the gas source, and APP as the acid source, and the intumescent system showed good flame retardancy for PU [109]. Hu et al. loaded molybdenum trioxide (MoO_3_), cuprous oxide (Cu_2_O), and cobalt (III) oxide (Co_3_O_4_) onto graphene sheets, and subsequently prepared PU/modified graphene composites with good flame-retardant and smoke-suppression properties by solution blending [110,111]. Nevertheless, the above methods of graphene and its derivatives as flame retardants were prepared by simple physical addition (solution blending, melt blending), but the blending method may not achieve complete layer separation of graphene, and because of the strong van der Waals forces between graphene layers and the weak interaction with PU, graphene tends to stack and agglomerate in the PU matrix, which is not conducive to the performance improvement of PU composites.

To weaken the intermolecular forces of graphene and prevent the agglomeration between layers, Hu et al. used lignosulfonate and cyclophosphazene flame retardants to modify graphene with non-covalent bonds, respectively. The modified graphene had good dispersibility in PU and improved the mechanical performance while enhancing the flame retardancy of the composites [112,113]. However, the non-covalent modification was unstable, and it could not improve the force between the graphene and PU matrix. Based on this, the modification of organic compounds on graphene by covalent bonding functionalization is an effective method. For instance, Huang et al. covalently grafted (3-aminopropyl)trimethoxysilane modified by Sb_2_O_3_ on the GO surface, then brominated it to elemental bromine on the GO surface, and finally reduced it to obtain flame-retardant-modified graphene Br–Sb_2_O_3_@RGO. By adding 10 wt% Br–Sb_2_O_3_@RGO to TPU, the peak heat release rate of the obtained composites decreased by 72% and the tensile strength increased by 37% [114]. Then, Huang et al. reported a P-N flame-retardant covalently modified graphene (PND-GO) structure and it significantly improved the flame retardancy and mechanical properties of PU composites prepared by solution blending [115].

Like other polymer/nano-composite systems, dispersion and interaction forces between graphene and PU matrix are the key to achieving high-performance PU nano-composites. Although the above-mentioned covalent bonding functionalization can effectively inhibit the agglomeration of graphene in composites, it can also greatly enhance the dispersion of graphene in the PU matrix and improve the interfacial interaction between them to a certain extent. However, the preparation of PU composites by blending does not achieve compatibility between graphene and the PU matrix at the molecular level, which is detrimental to the overall performance of composites. Hence, how to carry out surface modification and functionalization of graphene, design, and regulation of the interface matching between graphene and PU matrix is of great importance. Recently, flame-retardant-modified graphene was attached to polymers by covalent chemical bonding (e.g., epoxy resin [116], polyvinyl alcohol [117], polystyrene [106], polyethylene [118]), and polymer composites with excellent flame-retardant efficiency and overall performance were obtained due to the formation of strong covalent interfacial interactions. Until now, covalent grafting of graphene onto PU molecules to produce flame-retardant PU/graphene composites has not been reported so far.

### 3.4. Discussion about the Health Risks of Using Flame-Retardant Polyurethanes

Polyurethane is widely used in thermal insulation materials and household products [119], so the indoor air pollution caused by flame-retardant polyurethane has been paid attention to, especially as people spend most of their time indoors [120]. Additive flame retardants will not chemically bound to polyurethane products, so it is based on this to measure the concentration of flame retardants in indoor environment [119]. Recent studies have even reported the migration of flame retardants to artificial sweat through contact and dermal penetration (mediated by sweat) [121]. Among them, halogenated flame-retardants [122,123] and organophosphorus flame retardants [124] are mainly studied. Facing the health risks of flame-retardant polyurethane, we should have some basic knowledge: First, reduce the direct contact of flame-retardant polyurethane in the home environment; further improve the ventilation environment of gymnasium and other closed places; improve the personal protection of polyurethane recycling workers.

## 4. Applications of PU in Leather/Synthetic Leather

### 4.1. Applications of PU in Leather Fabrication

Leather is a natural polymer material with excellent hygienic properties, elegant appearance, comfortable feel, good physical and mechanical performance, and has been widely used in daily life and industrial products. From a manufacturing point of view, natural leather is made from raw animal pelts, which undergo a series of complex physical and chemical processes to produce a strong, durable leather product. Due to the dozens of processing steps, tannery processes require a large number of chemical reagents. Herein, PU is one of the most commonly used chemicals in the leather industry and its application can be traced back to the 1970s [125]. Currently, PU is primarily used as a finishing agent [126] and retanning filler [15].

#### 4.1.1. Retanning Filler

Currently, the application of PU retanning agent is mainly anionic WPU, which has excellent retanning, filling, and penetration effects in chrome-tanned leather. The tanning mechanism of anionic WPU can be briefly summarized as: during retanning, anionic WPU is coordinated and complexed with chromium in chrome-tanned leather through carboxyl groups, thereby increasing the molecular size of the chrome tanning agent and acting as an immobilizing agent for chromium [127]. Compared with acrylic acid-based retanning agents, PU also has characteristics of good levelling, dyeing assistance, and leather fiber dispersion [128].

#### 4.1.2. Finishing Agent

PU resin possesses characteristics of film forming, adhesion, flexibility, abrasion resistance, weather resistance, and dry rub resistance. At the same time, it has a comfortable and natural feel, and good compatibility with other resins, so it is an indispensable film-forming material in leather finishing (about 70–80%) [129,130]. The PU coating agent for leather is mainly WPU, which can be further divided into base coat (including patching agent), middle coat, and top coat WPU emulsions depending on the application.

In view of the importance of PU in the leather industry, the development of PU leather chemicals is paramount for the development of the leather industry. It is true that after nearly half a century of research and development, PU leather chemicals have made great progress, but progress is still needed in the development and application of high quality, multifunctional products.

### 4.2. Applications of PU in Synthetic Leather Fabrication

With the rapid development of the economy and the increasing population of the world, the demand for leather shoes, clothes, bags, and other products is increasing, and the demand for conventional natural leather resources exceeds the supply. To solve this conundrum, Dupont developed PU synthetic leather (brand name Corfam) in 1963, whose appearance and feel are similar to natural leather [131]. Over the past few decades, synthetic leather has gone through three generations of products, including PVC artificial leather, PU synthetic leather (PU leather), and ultra-fine fiber synthetic leather (microfiber leather). Synthetic leather has completed the transformation from low-grade to high-grade and simple imitation to high-level simulation. After years of research and development, the performance of synthetic leather has been comparable to or even surpassed that of natural leather, so the current market is strong and important for everyday life. From the manufacturing perspective, the development and application of PU synthetic leather and microfiber synthetic leather are inseparable from PU resin. PU is mainly used as the coating resin of PU leather and impregnated resin for microfiber leather substrates.

#### 4.2.1. PU Synthetic Leather Coating Resin

PU leather is a multilayer product formed by combining the coating of PU resin with substrates through dry lamination or wet processing. Specifically, in the dry PU leather production process, the PU slurry is directly coated on the substrate for production (direct scraping method); or the PU slurry is coated on the release paper, cured to obtain a uniform PU thin film, and then laminated to the substrate, and the leather paper is separated to afford leather products (transfer coating method). Nevertheless, the PU leather film produced by the dry method is dense and exhibits poor air permeability and hygiene properties [132]. For this reason, wet PU leather has also been developed by impregnating or coating DMF and PU slurry on the substrate, placing it in water, and then using the hydrophilicity of DMF, so that DMF is replaced by water in the coating. This causes the PU to solidify and form a continuous porous film, rendering a synthetic leather product that is comparable to natural leather in terms of air permeability, moisture permeability, feel, and appearance.

#### 4.2.2. Impregnation Resins for Ultra-Fine Fiber Synthetic Leather Substrates

The manufacturing of microfiber leather is usually based on a nonwoven fabric with a 3D network structure, which is impregnated with a solvent PU slurry, placed into a “H_2_O-DMF” coagulating bath, and then subjected to alkali reduction, grinding, dyeing, and finishing processes to obtain leather products. Since the structure of nonwoven fabric used in ultra-fine fiber is similar to that of bundled collagen fibers in natural leather, the solidified and reduced PU coating has an open microporous structure, which produces microfiber leather similar to natural leather in terms of microstructure and appearance. With better physical and mechanical performance and better air and moisture permeability, as well as better chemical resistance, hydrophobicity, and mildew resistance, it is one of the best alternatives to natural leather [133,134,135].

Nevertheless, synthetic leather production still suffers from serious pollution, urgent product structure adjustment, low value-added products, serious homogenization, and other defects. Currently, the production of synthetic leather is mainly based on solvated PU, and the amount of organic solvent accounts for more than 70% of the PU resin, which poses a great threat to the physical and mental health of workers and to the environment, as about 7–10% of the solvent cannot be recovered. With improved standards of living, the demand for high-performance, high-quality, and eco-synthetic leather continues to rise. Hence, synthetic leather will develop in the direction of ecological and functional synthetic leather that combines the advantages of good physical properties, multifunctionality, and eco-friendliness [133,136,137].

## 5. Studies of Flame Retardancy of Leather/Synthetic Leather

### 5.1. Flame-Retardant Leather

As mentioned earlier, natural leather products are made from animal pelt and then subjected to dozens of physical and chemical processes. In the leather industry, the use of a large number of organic polymers for retanning, fattening, and finishing has led to a decrease in the flame retardancy of leather [138]. The LOI of leather products without flame-retardant treatment is generally 23–27%, which is a combustible material, and the ignition time of leather is very long after ignition, accompanied by a lot of smoke and toxic gases. Therefore, the flammability of leather limits its application in forest fire protection equipment, aviation and automotive interiors, furniture leather, and other areas with high flame retardancy requirements.

Owing to the complicated processes in the leather industry, the flame retardancy of leather can be attributed to the integration of different tannery techniques and leather chemicals. Donmez et al. investigated the effects of tannery techniques on the flame retardancy of leather [139]. Previous studies discussed the effects of leather chemicals commonly used in retanning, fatliquoring, and finishing on the combustion performance of leather. The results demonstrated that compared with chrome-tanned leather, these stages will reduce flame retardancy of leather products [140,141]. Hence, it is not feasible to optimize and adjust the tannery techniques alone to obtain highly flame-retardant leather. Based on the optimized process, different commercial flame retardants were further used for leather retanning, and the flame retardancy of the treated leather was good, but there were problems such as poor feel and the roughness of the leather [142]. Therefore, Wang et al. synthesized various P-N flame-retardant retanning agents, and the retanning-treated leathers showed improved flame-retardant and sensory properties [143]. Melamine phosphates also imparted better flame retardancy to leather [144]. In recent years, the application of nanoparticles in flame-retardant leather has also been introduced. For instance, Olivares et al. added sodium-based MMT in the retanning stage, and the flame resistance of the leather products improved, but the mechanical performance decreased [145]. Then, Ma et al. mixed modified rapeseed oil and organic MMT to prepare a variety of modified rapeseed oil/modified MMT nano-composites, and the mechanical performance of the leather samples with its fatliquor was adequately maintained while the flame retardancy was improved [146].

The use of P-N flame retardants and nano-materials for leather processing can improve the flame retardancy of leather. However, most reported flame retardants still have inherent defects such as low flame-retardant efficiency, poor washing and wiping resistance, poor compatibility with leather fibers, and weak binding ability, which will undoubtedly affect the long-term flame retardancy of leather and its physical and mechanical performances. PU synthetic leather is primarily made of PU slurry, a fabric substrate (e.g., nonwoven, woven, knitted fabric), and related additives.

### 5.2. Flame-Retardant Synthetic Leather

Since PU, the substrate and most of the additives are flammable or combustible materials, the LOI of synthetic leather products is generally below 21% and defined as a combustible material, while hazardous gases and fumes are generated during the combustion process. In recent years, synthetic leather has been widely used in automobile interior decoration, clothing, furniture, luggage, aircraft and other fields, bringing comfort and convenience to everyday life, but its flammability characteristics can be a big safety hazard, threatening people’s lives and property. Hence, the research and development of flame-retardant synthetic leather has become a market trend. Overall, there are two ways to achieve flame retardancy of synthetic leather: one is to apply flame-retardant treatment to the substrate, and the other is to apply flame-retardant treatment to the PU resin.

#### 5.2.1. Flame-Retardant Substrate

The flame-retardant functional treatment of the substrate includes two methods: “reactive” and “additive”, i.e., the introduction of flame-retardant functional groups into the manufacturing process of the substrate to obtain flame-retardant fibers, or the use of flame retardants to treat the substrate. The former gives the substrate permanent flame retardancy, but the production cost is high, while the latter is simple and easy to implement. However, the flame-retardant of “additive” has a weak interaction with the substrate and greatly impacts the performance of the substrate. One “reactive” example, Liu et al. modified Nylon 66 with acrylamide by photo-grafting, and the LOI of the fabric was 26.2% when the photo-grafting rate was 32.5%, while providing good washing resistance in aqueous solution [147].

#### 5.2.2. Flame-Retardant PU Resin

The flame-retardant methods and research progress of PU have been described in detail, so we will not repeat them here. Notably, flame-retardant treatment of PU slurry is the most common way to provide flame retardancy to synthetic leather. In the practical production of synthetic leather, additive flame retardants such as Sb_2_O_3_, Al(OH)_3_, ammonium polyphosphate, and phosphate esters are blended with the PU slurry to enhance the flame retardancy of the leather products. In summary, the additive flame-retardant has disadvantages of a large amount of additive, poor washing resistance, and degraded mechanical properties of the synthetics, which all need to be urgently solved.

## 6. Conclusions and Perspectives

With its excellent overall performance, PU has been widely used in leather, synthetic leather, textiles, and other industries, bringing great optimization to human life. Nevertheless, conventional solvated PU has disadvantages such as organic solvent volatilization and pollution, which is contrary to the concept of sustainable development. Additionally, the flammability of PU also restricts its further utility in high-end applications. In view of the current situation and problems of the PU industry, the research pathways should focus on water-based, solvent-free PU resin, and functional, high-performance, and value-added PU products. In this review, the strategies to improve the flame-retardant performance of polyurethane were summarized from two aspects: additive and reactive flame retardants. In the meantime, the health risks of using flame-retardant polyurethane to human beings were discussed. More importantly, we have further discussed the application of polyurethane in the leather industry and the challenges it faces. For synthetic leather, research should be focused on the development of water-based, solvent-free PU and functional synthetic leather resins PU. Hence, it is of great theoretical significance and practical value to develop environmentally friendly flame-retardant PU that is compatible with ecological requirements and social development. Flame-retardant treatment of PU slurry is the most common way to provide flame-retardant synthetic leather. For the practical production of synthetic leather, it is important to develop efficient, non-toxic, durable, multifunctional flame retardants based on the characteristics of leather and leather chemicals so as to achieve high-performance, high-quality, functional, and diversified synthetic leather products.

The application prospects of flame-retardant polyurethane in leather industry are expansive. Although polyurethane synthetic leather cannot replace natural leather, it has been widely used in application scenarios such as car seats that are not in direct contact with people. The performance improvement of flame-retardant leather/synthetic leather depends on the advancement of basic research on flame-retardant polyurethane. In view of the problems mentioned above, we put forward some viewpoints and speculations to address them more comprehensively.

(1) Since eco-friendliness is the key to the application of polyurethane leather products, migration issues must be considered when designing flame retardants.

(2) The development of flame-retardant polyurethane should also consider its specific application, such as the natural feel and comfort in leather.

(3) Based on current research, P-N flame retardants and nano-material flame retardants are the development direction for obtaining high-quality leather products.

## Figures and Tables

**Figure 1 polymers-13-01730-f001:**
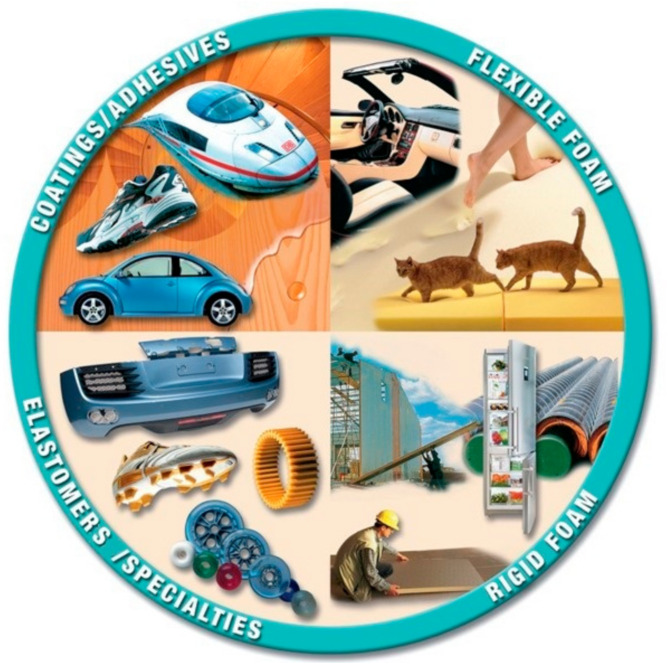
Related applications of PU materials [7]; used with permission from Angewandte Chemie International Edition, John Wiley and Sons, 2021.

**Figure 2 polymers-13-01730-f002:**
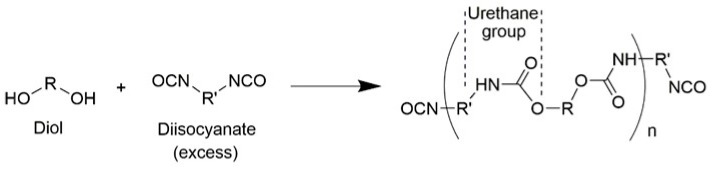
Schematic diagram of the PU synthesis mechanism.

**Figure 3 polymers-13-01730-f003:**
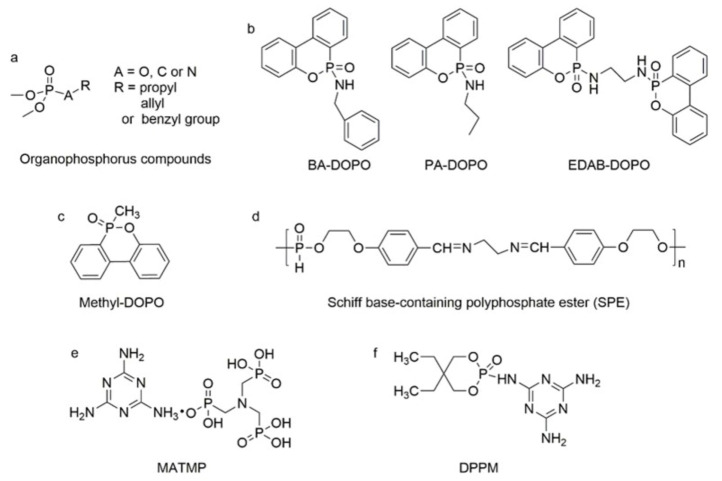
Chemical structures of phosphorus-containing and P–N flame retardants: (**a**) phosphonate, phosphate, and phosphoramidite flame-retardants; (**b**) DOPO phosphoramidite flame-retardants; (**c**) methylated DOPO flame-retardant; (**d**) Schiff base polyphosphate flame-retardant; (**e**) intumescent phosphoric acid based flame-retardant; (**f**) one-component intumescent phosphoramidite flame-retardant.

**Figure 4 polymers-13-01730-f004:**
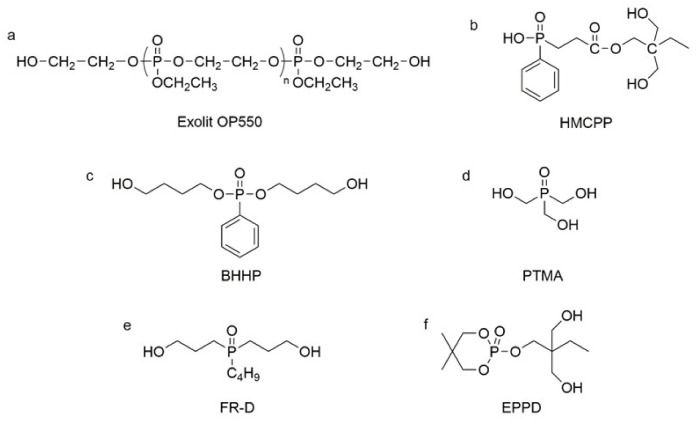
Chemical structures of phosphorus reactive flame retardants: (**a**) Clariant’s Exolit OP550 diol; (**b**,**c**) phosphonate diols; (**d**) trihydroxy phosphine oxide; (**e**) phosphine oxide-containing diols; (**f**) flame-retardant diol with phosphorus-containing side chains.

**Figure 5 polymers-13-01730-f005:**
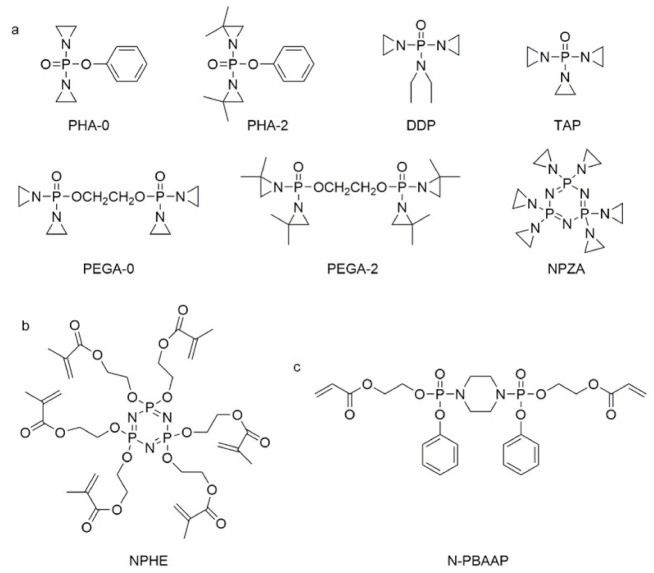
Chemical structures of P–N curing agents: (**a**) phosphorus-containing curing agents with aziridinyl groups; (**b**) methacryloyloxyethyl-terminated curing agent NPHE with cyclophphazene as the core from phosphonitrilic chloride trimer and hydroxyethylmethacrylate; (**c**) P–N curing agent N–PBAAP containing a bis(acryloyloxy) group.

**Figure 6 polymers-13-01730-f006:**
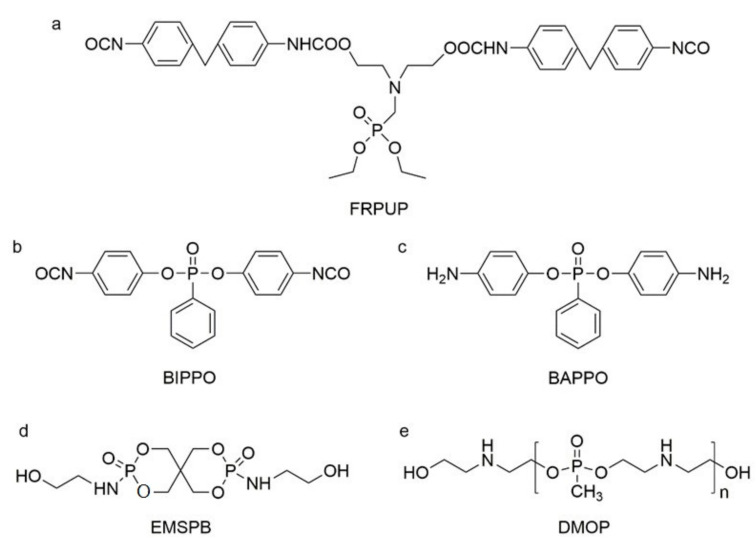
Chemical structures of P–N reactive flame-retardant: (**a**) flame-retardant diisocyanate synthesis by P–N diols reacted with methylene diphenyl diisocyanate; (**b**) P-N flame-retardant bis(4-isocyanatophenyl)phenylphosphine oxide; (**c**) bis(4-aminophenoxy)phenyl phosphine oxide; (**d**) phosphoramidite diols with a double spiro ring structure; (**e**) P–N macromolecular polyester diol using dimethyl methylphosphonate and diethanolamine.

**Table 1 polymers-13-01730-t001:** PU with additive flame retardants.

Name	Abbreviation	Molecular Weight (g/mol)	Chemical Structure
Aluminum hydroxide	ATH [24]	78	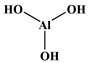
Magnesium hydroxide	MTH [25]	58.3	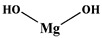
Ammonium polyphosphate	APP [23]	-	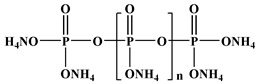
Tris(2-chloroethyl)phosphite	TCEP [21]	285.5	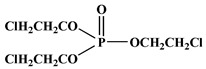
Tris(chlorisopropyl)phosphate	TCPP [21]	327.5	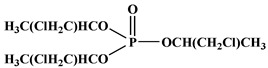
Tris(1,3-dichloro-2-propyl) Phosphate	TDCP [21]	430.9	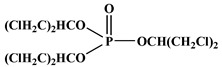
CR-505 flame-retardant	CR-505 [26]	516	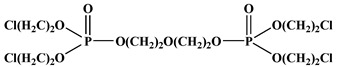
Tri(2,3-dibromopropyl) phosphate	TDBPP [27]	697.5	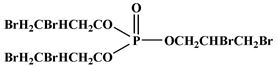
Dimethyl methylphosphonate	DMMP [28]	124.1	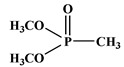
Decabromodiphenyl ether	DBDPO [29]	959.2	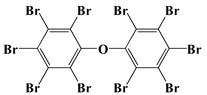
Melamine	MA [22]	126.1	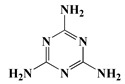
Melamine cyanurate	MC [30]	255.2	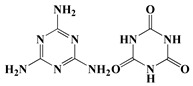
Melamine pyrophosphate	MPP [31]	304.1	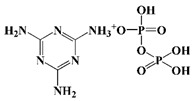

**Table 2 polymers-13-01730-t002:** Chemical structures of flame-retarded polyols with built P and N atoms.

Reference	Chemical Structure	Abbreviation	Content of Flame Retardant %	LOI Values %	UL-94 Ratings
[81]	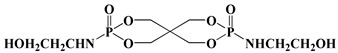	EMSPB	25	27.5	V-0
[82]	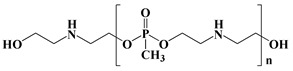	DMOP	6.3	22.1	-
[83]	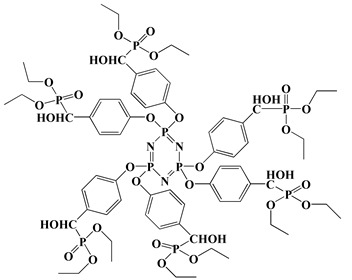	HPHPCP	20	26	HF-1
[84]	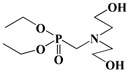	BHAPE	6	28.1	-
[85]	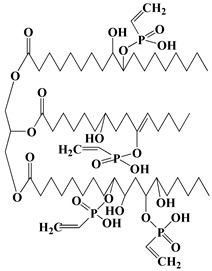	PoIP	-	26.4	-

**Table 3 polymers-13-01730-t003:** Chemical structures of typical P–N reactive flame retardants and performance of flame-retardant WPU.

Reference	Chemical Structure	Abbreviation	Amount (%)	Flame Retardancy
[86]	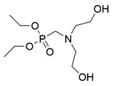	Fyrol-6	19.0	LOI is 29%, UL-94 V-2 rating
[87]	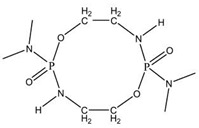	ODDP	15.0	LOI is 30.6%, UL-94 V-0 rating
[88]	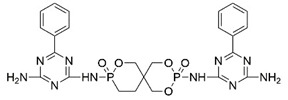	BSBP	8.0	LOI is 27.3%, UL-94 V-0 rating
[89]	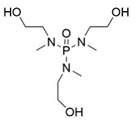	FROH	8.2	LOI is 33%, UL-94 V-0 rating
[90]	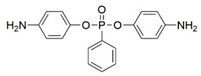	BPPO	-	LOI is 30.1%, 57.1% reduction in peak heat release rate
[91]	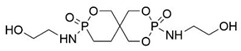	PDNP	9.0	LOI is 26%

## Data Availability

Not applicable.

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
