# Peer review of "Preparation of Flame-Retardant Polyurethane and Its Applications in the Leather Industry"

_polymers, 2021, doi:10.3390/polym13111730_

Round 1

Reviewer 1 Report

Table 1 - please provide the chemical formulas in the same way for all compounds.

Table 2 - caption is missing.

Generally the Authors did a great job summarizing the literature about the flame retardancy of PUs. However, I miss couple of things:

  • Tables which will summarize and compare the effect of various additives, e.g. their effect on LOI or PHR or THR or other parameters,
  • what about flame retarded polyols with built P and N atoms,
  • What about expandable graphite, which is quite popular?
  • Describe better the mechanisms of actions of particular flame retardants.

Reviewer 2 Report

Dear authors,

As one of the selected reviewers, I read your manuscript carefully. While your research has addressed an important subject, however, I observed some issues that must be addressed in a newer version.

  1. What are the health risks of using these flame-retardant polyurethanes, add a description about it.
  2. Add respective citations in Table 1.
  3. What are the prospects, add a separate section about? 
  4. Also, no research limitation or research gap of this research direction is explained.
  5. Describe what dimensions of the problem have been found by you to explore further.
  6. I suggest you add a discussion before a concrete conclusion.
  7. Check punctuations carefully, for example, see additional unnecessary full stop at line 323.
  8. There are too many self-citations in the manuscript, which should be reduced.
  9. Check subscripts, for example, see line #40.

I look forward to receiving the revision and reviewing the newer version.
Best of luck"

Round 2

Reviewer 1 Report

Everything in order after corrections

Reviewer 2 Report

The authors have improved the manuscript as per the reviewer's guidance; therefore, the manuscript might be considered for publication in its present form.